# Bio-Inspired Conceptual Mechanical Design and Control of a New Human Upper Limb Exoskeleton

Narek Zakaryan *, Mikayel Harutyunyan and Yuri Sargsyan 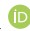





Department of Mechanics and Machine Science, National Polytechnic University of Armenia, Yerevan 0009,
Armenia; mharut@seua.am (M.H.); yusarg@seua.am (Y.S.)
* Correspondence: n.zaqaryan@polytechnic.am; Tel.: +37-494-355-234

**Abstract:** Safe operation, energy efficiency, versatility and kinematic compatibility are the most
important aspects in the design of rehabilitation exoskeletons. This paper focuses on the conceptual
bio-inspired mechanical design and equilibrium point control (EP) of a new human upper limb
exoskeleton. Considering the upper limb as a multi-muscle redundant system, a similar over-actuated
but cable-driven mechatronic system is developed to imitate upper limb motor functions. Additional
torque adjusting systems at the joints allow users to lift light weights necessary for activities of daily
living (ADL) without increasing electric motor powers of the device. A theoretical model of the "ideal"
artificial muscle exoskeleton is also developed using Hill's natural muscle model. Optimal design
parameters of the exoskeleton are defined using the differential evolution (DE) method as a technique
of a multi-objective optimization. The proposed cable-driven exoskeleton was then fabricated and
tested on a healthy subject. Results showed that the proposed system fulfils the desired aim properly,
so that it can be utilized in the design of rehabilitation robots. Further studies may include a spatial
mechanism design, which is especially important for the shoulder rehabilitation, and development
of reinforcement learning control algorithms to provide more efficient rehabilitation treatment.

**Keywords:** upper limb rehabilitation; bio-inspired exoskeleton; cable-driven system; over-actuation;
Hill's model; EP control; torque adjusting mechanism; differential evolution

## 1. Introduction

The nowadays fast developing robotics technology promises to improve human upper-
limb functionalities required for performing ADL [1–5]. The following technical challenges
urgently need to be studied:

- Kinematic compatibility,
- Safety,
- Control strategy.

The bio-inspired design (bio-inspiration) can offer the best solutions in rehabilitation
robotics, particularly in wearable robotics. They lead to a similarly efficient and effective
robotic system design. The main characteristics of a bio-inspired design include the
redundant actuation, flexible actuators, flexible links and joints, and an appropriate bio-
inspired control. Successful examples of upper limb and hand bio-inspired exoskeletons
have been presented recently by Ning Li et al. and Ong et al. [6,7]. They used tension lines,
cables and flexible bands to simulate muscles and muscle tendons of the human upper
limb. Test results confirmed that a natural-like joint motion range and a trajectory curve
are provided; thus, the effectiveness of the bio-inspired design is proven. These studies
are devoted to bio-inspired mechanical design, i.e., ensuring kinematic compatibility, but
no appropriate control systems have been developed, and the capabilities of these devices
have not been evaluated when used in human ADL.

The well-known studies of bio-inspired exoskeletons mainly refer to cable-driven
exoskeletons, where Bowden cables are widely used [8–11]. Many leading scientists have

---

studied cable-driven exoskeletons for stroke patients [12–15]. This innovative design increases the robot safety and effectiveness. It is shown that the remote control of arm movements helps to achieve a light weight and low inertia design properties. Besides, the cable-driven architecture can eliminate possible joint misalignment between the human upper limb and exoskeleton, thus reducing the chances of injuries for the patient during robotic rehabilitation [16,17]. However, bio-inspired control methods are proposed in those studies, although they are not derived from the control principles of natural biological systems, and they do not take into account that the bio-inspired control mostly depends on the nonlinear properties of natural muscles.

The basic notions of the EP control theory are related to the multi-muscle and multi-degrees of freedom redundancy [18–22]. As is known, the shift of equilibrium positions is ensured in the result of the organism–environment interaction; therefore, a new generation of a robot–human interaction can play a crucial role. According to the investigations of Feldman et al., the notion of EP characterizes the equilibrium state in terms of equilibrium positions of body segments and the muscle torques at these positions as two-dimensional vectors [23]. However, each EP is associated with a static (steady) state of the system, the notion of EP shifts is essentially dynamical, and its presence essentially influences the dynamics of motor behavior. In fact, the EP does not conflict with dynamic systems theory; rather, it complements it. It is a specific form of dynamic systems theory.

Spiers A. et al. have studied muscle models as well, highlighting their role in bio-inspired robots [24]. A popular method of characterizing muscle motion is the Hill model with its variations [25–27]. Emerged actuator technologies such as electroactive polymers (EAPs) are promising alternatives for natural muscles, but their characteristics are not satisfactory yet, and they are in the stage of research and improvement [28].

In this paper, a conceptual model of a cable-driven over-actuated upper limb rehabilitation device with mechanical variable torque mechanisms at the joints and EP control, providing safety and human-like motions, is described. A universal artificial muscle-driven exoskeleton using Hill's natural muscle model as an actuator is also considered, and results are compared. The proposed cable-driven exoskeleton is prepared and tested, which ensured light weight, flexibility, kinematic compatibility of the device and, therefore is safe and comfortable for the user, confirming the theoretical results, and creating a basis for further development. Our eventual goal is the development of a theoretical model of an upper limb exoskeleton, which can be used as a guide to evaluate the performance of the proposed cable-driven device, as well as other similar artificial muscle-devices, and finally, the use of the architecture described herein to create robots which will be suitable for use in domestic and rehabilitation environments, where they will perform tasks that are normally performed in human ADL.

## 2. Conceptual Mechanical Design of the Exoskeleton

The human upper limb is a system with great mobility and muscle redundancy, which can be modeled by an approximation biomechanical model consisting of only 7 DoFs (shoulder-3 DoF, elbow-2DoF, wrist-2 DoF). The total number of upper limb muscles is about 30, among which there are biarticular muscles, complicating the system even more. Many of the studied human motion dynamic optimization methods have included models of muscle forces and dynamics [29,30]. The fact that several muscles are involved in the same joint movement was also considered [31]. The movement of a joint is a combination of these effects (synergy). For example, elbow flexion (i.e., folding of the elbow so that the angle between the arm and forearm decreases) has various levels of torque contributions from four separate muscles in the arm (the brachialis, biceps brachii, brachioradialis and pronator teres muscles).

It is proposed to simplify the system in this work, by reducing the degrees of freedom of the biomechanical model and considering its movements in only the sagittal plane. Based on the agonist–antagonist structure of the natural muscular system of human limbs, a new concept of a device, actuated by a cable-driven system, is presented (Figure 1).

Bi-articular cables (actuation of two joints with single cable) are also included, which makes the system more complex, but closer to the human muscular system. In addition, it has variable torque mechanisms at the joints, with $h_1, h_1', h_2, h_2', h_3, h_3'$ variable lever arms.

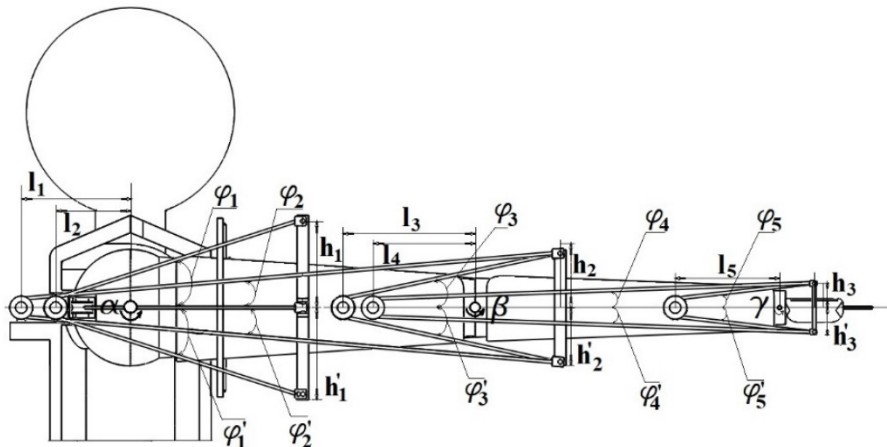

**Figure 1.** The upper limb bio-inspired exoskeleton (sagittal plane, conceptual design).

Cable tensions can be represented by generalized coordinates in order to use Lagrange's method. Suppose that $c_j$ are position vectors of the connection point of the jth cable to the exoskeleton expressed in the Cartesian frame. The generalized forces ($Q_i^t$) of the system can be expressed in cable forces as follows:

$$Q_i^t = \sum_{j=1}^{10} \left( F_j w_j \cdot \frac{\partial c_j}{\partial q_i} \right), i = 1 \ldots 3, \tag{1}$$

where $w_j$, $c_j$ are unit direction vectors of the jth cable and the corresponding moment arm on the end-effector, respectively, $q_i (i = 1 \ldots 3)$ or $\alpha, \beta, \gamma$ are generalized coordinates.

Equilibrium equations of the system can be written as follows:

$$KF = M_f \tag{2}$$

where F is the matrix of cable tensions:

$$F = \left| \begin{array}{cccc} F_1 & F_2 & \ldots F_{10} \end{array} \right|^T. \tag{3}$$

$M_f$ includes all the external forces and represents the following matrix:

$$M_f = \left| \begin{array}{ccc} \frac{d}{dt}\left(\frac{\partial L}{\partial \dot{\alpha}}\right) - \frac{\partial L}{\partial \alpha} - Q_{g_1} & \frac{d}{dt}\left(\frac{\partial L}{\partial \dot{\beta}}\right) - \frac{\partial L}{\partial \beta} - Q_{g_2} & \frac{d}{dt}\left(\frac{\partial L}{\partial \dot{\gamma}}\right) - \frac{\partial L}{\partial \gamma} - Q_{g_3} \end{array} \right|^T \tag{4}$$

where L is the Lagrangian, $Q_{g_1}, Q_{g_2}, Q_{g_3}$ are gravity forces, K is the structural matrix, including unit direction vectors $w_j$, $c_j$, L is determined by the difference between kinetic and potential energies of the system.

The lever arms ($a_i$) can be defined by the following diagram (Figure 2):

$$\sin \varphi_i = \frac{a_i}{l_i + b_i} = \frac{r_i}{b_i} \Rightarrow a_i = l_i \sin \varphi_i + r_i, i = 1 \ldots 5$$
$$\text{or } a_i = l_i \frac{h_i}{AF} + r_i = l_i \frac{h_i}{\sqrt{h_i^2 + AG^2}} + r_i \tag{5}$$

where the distances AG and AF can be chosen based on the sizes of the human upper limb.

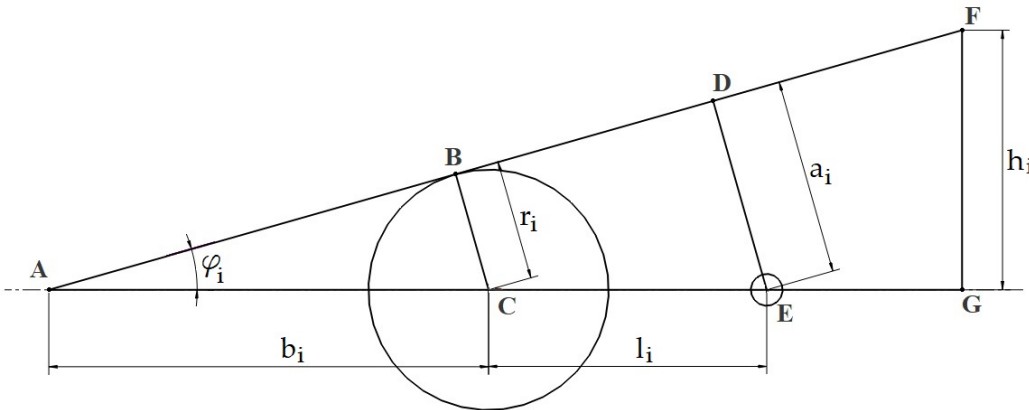

**Figure 2.** Geometrical parameters of the exoskeleton actuation system.

Equations (1)–(5) allow to establish the relation between cable tensions, geometrical parameters of their attachments, radiuses of pulleys and variable lever arms.

Let us now consider the structure and operation of the actuation system (Figure 3).

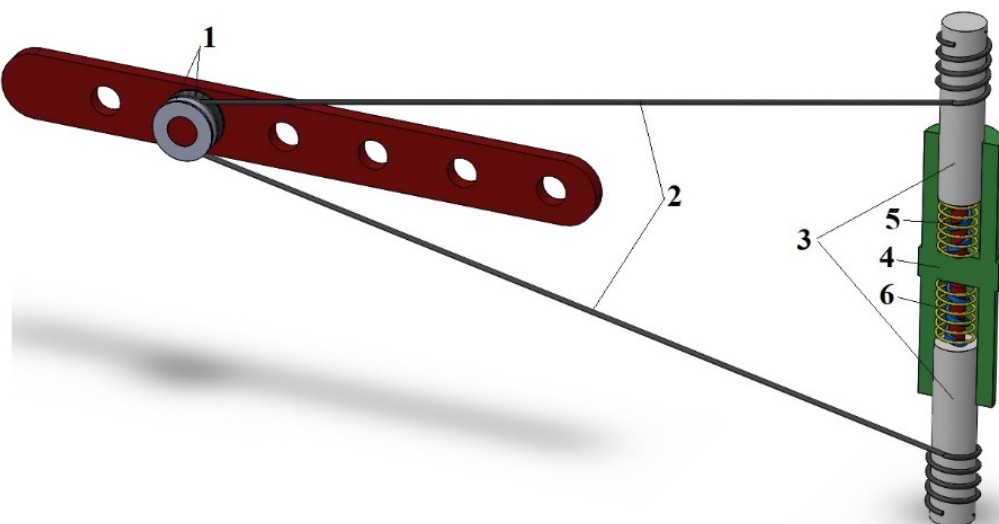

**Figure 3.** Conceptual design of the exoskeleton actuation module.

The device works as follows: the agonist and antagonist actuators (2) are connected to the corresponding levers (3) of the mechanism pulleys (1). The levers form screw pairs with the links (5) and are mounted in the cylinder (4) by attaching to the spring (6). When the values of tensile forces of the actuators are exceeding a certain value during the operation of the device, the levers start to rotate towards the cylinder in the anti-clockwise direction, the lever arms of the cable tensile forces are increasing, and consequently, the torques at the joints are also increasing. When the load is decreasing or partly restored muscles participate in the movement, the spring restores the lever initial length, reducing the torque.

Thus, a cable-driven bioinspired exoskeleton was developed, and a mechanism providing variable torque in the joints was additionally installed to regulate the robot's lifting capability. The next step is to determine the optimal values of the device design parameters. However, the analysis of the torque adjusting mechanism will be considered in our future studies.

## 3. Design Optimization Using Differential Evolution Method

Differential evolution (DE) is an evolutionary algorithm, which uses the difference of solution vectors to create new candidate solutions. DE was originally proposed by

Rainer Storn and Kenneth Price in 1997 [32]. DE is a very simple, yet very powerful and useful algorithm, and can be used to deal with a wide variety of optimization problems. In this paper, we design a bio-inspired rehabilitation exoskeleton using multi-objective DE algorithm. According to the proposed control strategy (EP hypothesis), the system's movement consists of balanced positions, which is treated as a constraint. Thus, the multi-objective optimization considers the following three objectives to minimize:

(1)    The total mass of the device,
(2)    The maximal magnitudes of cable tensions,
(3)    The maximal difference between magnitudes of agonist-antagonist cable tensions.

The first goal is dedicated to reducing the weights of arm, forearm and hand segments of the exoskeleton ($m_{up}$, $m_f$, $m_h$), and the main components of the cable-driven mechanism (pulleys ($m_{p_i}$) and electric motors ($m_{e_i}$)), in order to design a lightweight and wearable rehabilitation device. The total mass of the exoskeleton can be defined as follows:

$$M_{exos} = m_{up} + m_f + m_h + \sum_{i=1}^{10}\left(m_{p_i} + m_{e_i}\right), \tag{6}$$

The second objective concerns the reduction of cable tensions, which will lead to the use of smaller electric motors and, consequently, energy saving. In multi-body cable-driven mechanisms, the effect of cables is modeled as point forces applied to the links, i.e., inertia and elasticity of the cables are ignored. For simplicity, it is accepted that the radii of the agonist and antagonist pulleys are equal.

Achieving the third goal will increase the rigidity of joints, will make the device safer, increase the accuracy of upper limb movements, but as a result, energy consumption will be increased. It can be expressed as follows:

$$\min_t \left| |F_i| - |F'_i| \right|, i = 1 \ldots 10. \tag{7}$$

According to the design objectives, the following design variables have been selected:

- the masses of segments of the exoskeleton ($m_{up}$, $m_f$, $m_h$), pulleys ($m_{P_1}$, $m_{P_2}$, $m_{P_3}$, $m_{P_4}$, $m_{P_5}$, $m'_{P_1}$, $m'_{P_2}$, $m'_{P_3}$, $m'_{P_4}$, $m'_{P_5}$) and electric motors ($m_{e_1}$, $m_{e_2}$, $m_{e_3}$, $m_{e_4}$, $m_{e_5}$, $m'_{e_1}$, $m'_{e_2}$, $m'_{e_3}$, $m'_{e_4}$, $m'_{e_5}$),
- positions of pulleys installation ($l_1$, $l_2$, $l_4$, $l_5$, $l_7$), radii of pulleys ($r_1$, $r_2$, $r_3$, $r_4$, $r_5$), and cable connection angles ($\varphi_1$, $\varphi_2$, $\varphi_3$, $\varphi_4$, $\varphi_5$, $\varphi'_1$, $\varphi'_2$, $\varphi'_3$, $\varphi'_4$, $\varphi'_5$),
- cable tensions ($F_1$, $F_2$, $F_3$, $F_4$, $F_5$, $F'_1$, $F'_2$, $F'_3$, $F'_4$, $F'_5$).

Thus, the vector of design variables, R, will be defined as follows:

$$R = \left[ m_{up}, m_f, m_h, m_{p_i}, m_{e_i}, r_i, \varphi_i, \varphi'_i, F_i, F'_i \right], i = 1, 2, \ldots, 5. \tag{8}$$

The next important step is to specify the range of possible values of the design parameters (Table 1) that meet the requirements of wearable rehabilitation devices.

**Table 1.** Bounds of design variables.

| Variables | Range | Units |
|:---:|:---:|:---:|
| $m_{up}$ | [1, 5] | kg |
| $m_f$ | [0.5, 4] | kg |
| $m_h$ | [0.1, 0.5] | kg |
| $m_{p_i}^j$ | [0.1, 0.5] | kg |
| $m_{e_i}^j$ | [0.5, 1] | kg |
| $r_i^j$ | [0.01, 0.1] | m |
| $\varphi_i^j, \varphi_j^{'j}$ | [5, 90] | deg |
| $F_i^j, F_i^{'j}$ | [1, 50] | N |

An algorithm is described to apply according to the simple evolutionary algorithm [33–35].

In simple DE, an initial random population consisting of NP vectors is randomly generated according to a uniform distribution within the lower and upper boundaries $(a_{jL}, a_{jU})$. After initialization, these individuals are evolved by mutation and crossover to generate a trial vector. A comparison between the parent and its trial vector is then done to select the vector which should survive to the next generation. To start the optimization process, an initial population must be created. Each jth component $(j = 1, 2, \ldots, D)$ of the ith individuals $(i = 1, 2, \ldots, NP)$ is obtained as follows:

$$a_{ji} = a_{jL} + rand(0, 1)(a_{jU} - a_{jL}), \tag{9}$$

where $rand(0, 1)$ returns a random number in $[0, 1]$, D is the number of variables to be optimized.

A mutant vector $\mu_i$ is generated according to the following:

$$\mu_i = a_{n_1} + F(a_{n_2} - a_{n_3}), \tag{10}$$

where $n_1, n_2, n_3 \in \{1, 2, \ldots, NP\}$, are randomly chosen indices, $F = [0, 2]$ is a real number to control the amplification of the difference vector. The crossover rate $(CR \in [0, 1])$ is also important and introduced to control the number of components inherited from the mutant vector.

MATLAB codes of the described above algorithm are presented in Appendix A.

## 4. Control Strategy Analysis: EP Control

Wearable robots are designed to interact with humans. Consequently, they must be designed for intrinsic safety and should repeat human natural movements. The key design features, promoting intrinsic safety, are the following:

- mechanical compliance to accommodate interactions,
- light weight to minimize kinetic energy,
- bio-inspired control strategy.

Both joint stiffness and equilibrium position of each joint in the robot can be controlled using the MIMO control system [36]. The errors in joint stiffness k and equilibrium position $\theta_{EP}$, along with the actual angular position, are transferred into a decoupling block (Figure 4), which uses the partial derivatives of joint stiffness $\partial K_i / \partial F_i$ and joint EP $\partial \theta_{EP} / \partial F_i$ $(i = 1, 2, 3)$ with respect to cable tension to transform stiffness and EP to error in tension for each cable.

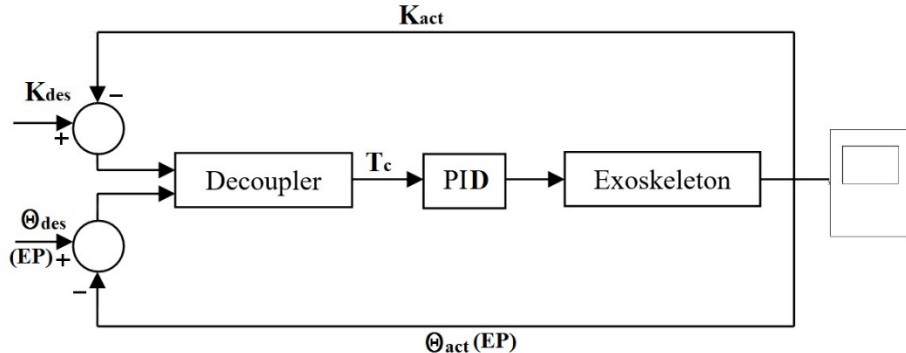

**Figure 4.** EP control architecture of the exoskeleton.

In order to obtain these derivatives, we note that torque in each joint is given by

$$\tau = K(\theta - \theta_{EP}) = 1 \sum \overline{F}. \tag{11}$$

The PID controller was tuned using MSC ADAMS control toolkit methods.

The hand position can be determined analytically, in terms of two dimensional homogeneous transformations—rotational and translational:

$$P_H = R(\alpha)T(L_1)R(\beta)T(L_2)R(\gamma)T(L_3). \tag{12}$$

By multiplying all of the matrices, we get

$$P_H = \begin{pmatrix} C_1 & -S_1 & L_2C_2 + L_1C_3 + L_3C_1 \\ S_1 & C_1 & L_2S_2 + L_1S_3 + L_3S_1 \\ 0 & 0 & 1 \end{pmatrix}, \tag{13}$$

where $C_1 = \cos(\alpha + \beta + \gamma)$, $C_2 = \cos(\alpha + \beta)$, $C_3 = \cos\alpha$, $S_1 = \sin(\alpha + \beta + \gamma)$, $S_2 = \sin(\alpha + \beta)$, $S_3 = \sin\alpha$.

The joint stiffnesses are controlled by the agonist–antagonist artificial muscular system with considerable accuracy, which is very important for the rehabilitation/assistive device; therefore, consideration of frictional forces in dynamic equations of the system is also important:

$$\frac{d}{dt}\left(\frac{\partial T}{\partial \dot{\theta}_i}\right) - \frac{\partial T}{\partial \theta_i} + \frac{\partial V}{\partial \theta_i} + \frac{\partial F_R}{\partial \dot{\theta}_i} = \tau_i, i = 1, 2, 3, \tag{14}$$

where $\theta = \begin{bmatrix} \alpha & \beta & \gamma \end{bmatrix}^T$, $\tau = \begin{bmatrix} \tau_1 & \tau_2 & \tau_3 \end{bmatrix}^T$, T-system kinetic energy, V-system potential energy, $F_R$-energy dissipation by joint frictions, which can be defined by the following expressions:

$$T = \frac{1}{2}\sum_{i=1}^{3} J_i\dot{\theta}_i^2 + m_i\left[\frac{d}{dt}\left(\sum_{k=i-2}^{i-1}(L_k\sin\theta_k + a_i\sin\theta_i)\right)\right]^2, \tag{15}$$

$$V = \sum_{i=1}^{3} m_ig\left(a_i\cos\theta_i + \sum_{k=i-2}^{i-1} L_k\cos\theta_k\right), \tag{16}$$

$$F_R = \frac{1}{2}\sum_{i=1}^{3} c_i(\theta_i - \theta_{i-1})^2, \tag{17}$$

where $L_i$, $a_i$, $m_i$, $J_i$ and $\theta_i$ are link lengths, distances from the centroid to corresponding axes, masses, moments of inertia computed by means of three link centroids, and angles of deviation from vertical direction are measured clockwise, respectively, $i = 1, 2, 3$ are indexes of three links, while the frictional torque coefficients of three joints are denoted as $c_1$, $c_2$ and $c_3$.

Kinematic modeling and new control principle based on the EP hypothesis can be implemented by using MSC ADAMS software with the following assumptions: the whole range of upper limb segments motion is divided into 26 equilibrium position subranges (Figure 5a), using flexion/extension ranges of upper limb joints (shoulder: $(-180 \ldots +80)$, elbow: $(-10 \ldots +145)$, wrist: $(-90 \ldots +70)$), and the movement is considered in the sagittal plane, where flexion/extension rotations of the shoulder, elbow and wrist joints are only possible.

The control system works as follows: desired angles of joints rotations, and the durations of breaks or balancing are prescribed, then the algorithm finds the necessary torques, which can provide the given motion.

Computer simulations (Figure 5b) are performed, according to the positions shown in Figure 5a.

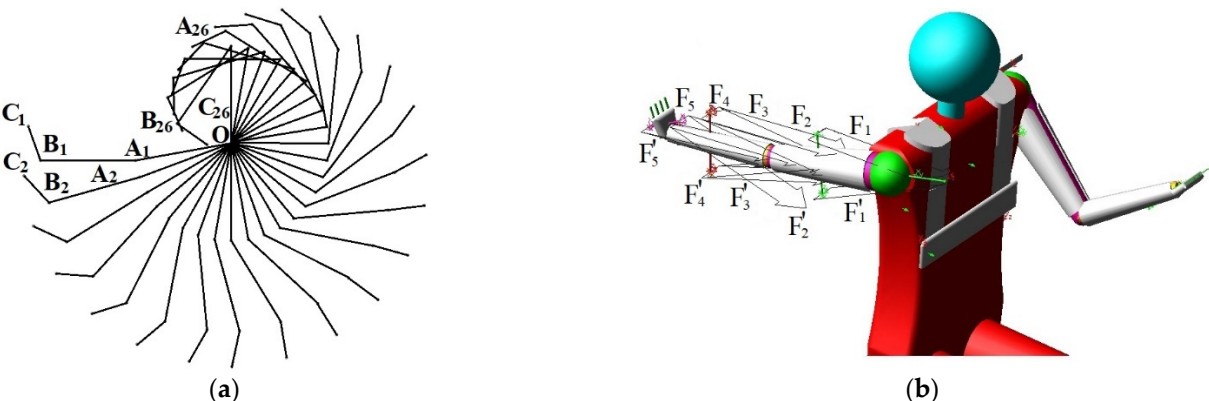

(a)                                                (b)

**Figure 5.** (**a**) 26 balancing postures of the upper limb (Arm-$OA_i$, Forearm-$A_iB_i$, Hand-$B_iC_i$, $i = 1, 2, \ldots, 26$), (**b**) MSC ADAMS simulation of the over-actuated upper limb model (Wrist: STEP (time, 0, 0, 1, $-6.15$ d)+ . . . + (26th)STEP (time, 0, 0, 1, $-6.15$ d), Elbow: STEP (time, 0, 0, 1, $-5.96$ d)+ . . . + (26th)STEP (time, 0, 0, 1, $-5.96$ d), Shoulder: STEP (time, 0, 0, 1, $-10$ d)+ . . . + (26th)STEP (time, 0, 0, 1, $-10$ d).

## 5. Artificial Muscle Model and System

In this part, the cables are replaced by artificial muscles, of which the properties are similar to the properties of natural muscles. For this purpose, Hill's natural muscle model is studied, and is used to represent all necessary properties of the proposed artificial muscle [37]. The series elastic element models the behaviour of the tendon and the connective tissues. The parallel elastic element reflects the resistance of the muscle to passive stretching, while the damper models the dynamic resistance to movement, which is speed dependent. There is only one active element that models the contraction of the fibres; this is a force generator.

Based on the Hills natural muscle model, a new model of an artificial muscle is designed. For this purpose, the series elastic element is presented by two triangular components to ensure the stability of the system (Figure 6a).

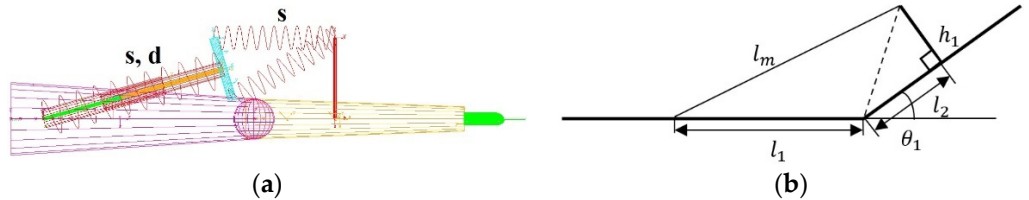

(a)                                                (b)

**Figure 6.** (**a**) A 1 DOF manipulator system actuated by Hill's model artificial muscle, (**b**) the kinematic scheme.

Computer experiments of this 1 DoF manipulator system (only the elbow joint is actuated) are implemented by means of MSC ADAMS software.

Kinematics and dynamics of a single link musculoskeletal structure are presented by Shukor et al. [38], where the lengths of actuators are determined by a joint rotation angle, expressed by coordinates of actuator connection points. In this case, the total length of the actuator is the sum of current lengths of the muscle model series and parallel elastic elements. In addition, the connection to the second link is made rigidly by means of additional $h_1$ length link and vertically attached to it. Therefore, the total muscle length ($l_m$) can be determined using the kinematic scheme (Figure 6b).

According to the generalized Pythagorean theorem:

$$l_m = \sqrt{l_1^2 + h_1^2 + l_2^2 - 2l_1\sqrt{h_1^2 + l_2^2}\cos\left(\theta_1 + \arctan\frac{h_1}{l_2}\right)},$$
$$l_m = x_1 + x_2 \sin\vartheta, \tag{18}$$

where $\vartheta$—connection angle of series elastic elements, $x_1$, $x_2$—deformations of parallel and series elastic elements, respectively.

Since we have a muscle model, the well-known formulas for muscle contraction can be used to get an idea of how the model works. As is known, the muscle contraction force is determined by the rate of contraction ($V_m$), the length of the muscle ($l_m$), and activation ($a(t) \in [0, 1]$) [39].

$$F_c = F_{max}a(t)f(l_m)f(V_m), \tag{19}$$

where: $f(l_m)$—force-length relation of the muscle, $f(V_m)$—force-velocity relation of the muscle; $F_{max}$—maximal isometric force at optimum muscle fiber length and zero velocity.

Muscle length and velocity can be chosen as state space variables: $X = \begin{bmatrix} l_m & \dot{l}_m \end{bmatrix}^T$.

The contraction force (Figure 7a) and the rate of contraction (Figure 7b) are obtained from the motion study of the upper limb from the horizontal position to the maximum flexed position. The motion duration is 2 s. The average speed: 0.12 m/s: The stiffness and damper coefficients of parallel elastic element and parallel damper are determined by acceptable maximum values of force generator through computer simulations: 0.5 N/mm and 0.3 N·s/mm respectively.

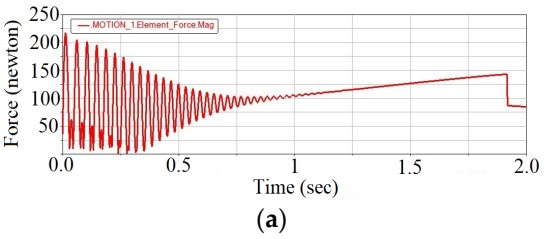
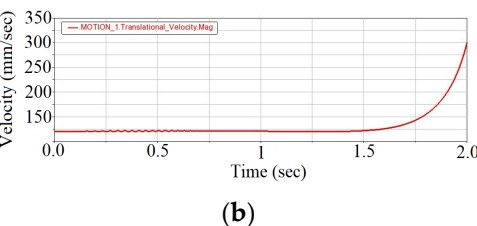

**(a)** **(b)**

**Figure 7.** Motion study of the 1 DOF manipulator system: (**a**) the contraction force, (**b**) the rate of contraction.

Substituting obtained values from the computer simulation to the (19), we can calculate $F_{max}$, which represents the necessary preloads of springs. On the reducing of vibrations and maximal forces should be dedicated the optimal design: the determination of optimal values of elastic elements, dampers and system sizes.

There exist similar studies of generalized models with one link and multiple actuators [37], with two links and four actuators [38], with three links and nine actuators [39]. However, there is no generalized study of the model with 3 links and multiple natural muscle-like actuators.

Finally, computer simulations of the conceptual model of the upper limb-exoskeleton system actuated by artificial muscles with properties similar to natural muscles (Hill's model) can be provided (Figure 8).

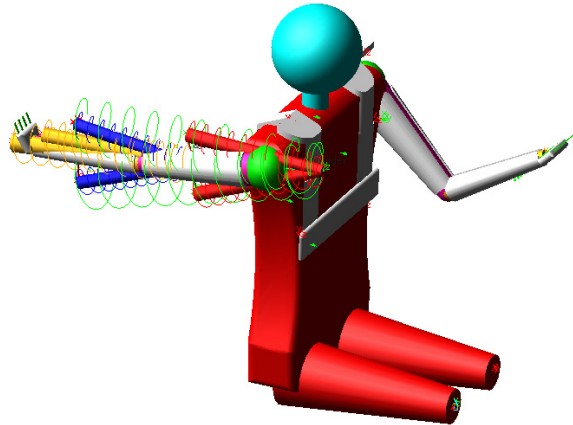

**Figure 8.** MSC ADAMS model of the upper-limb exoskeleton system with Hill's type artificial muscles.

Obtained values of the forces can be used to determine the stiffness and damping coefficients of the natural muscle model components. Results of the computer modelling are presented in the Table 2. For the sake of simplicity, stiffness of parallel and series elastic elements are considered equivalent.

**Table 2.** Mechanical properties of Hill's model components.

| Stiffness, N/m | $s_1$ | $s_1'$ | $s_2$ | $s_2'$ | $s_3$ | $s_3'$ | $s_4$ | $s_4'$ | $s_5$ | $s_5'$ |
|---|---|---|---|---|---|---|---|---|---|---|
| | 300 | 300 | 140 | 100 | 800 | 270 | 750 | 220 | 70 | 80 |
| Damping, N·s/m | $d_1$ | $d_1'$ | $d_2$ | $d_2'$ | $d_3$ | $d_3'$ | $d_4$ | $d_4'$ | $d_5$ | $d_5'$ |
| | 95 | 95 | 95 | 95 | 40 | 40 | 40 | 40 | 17 | 17 |

## 6. Experimental Validation

The proposed exoskeleton is fabricated (Figure 9) and tested on a healthy subject (see Video S1, Supplementary Materials).

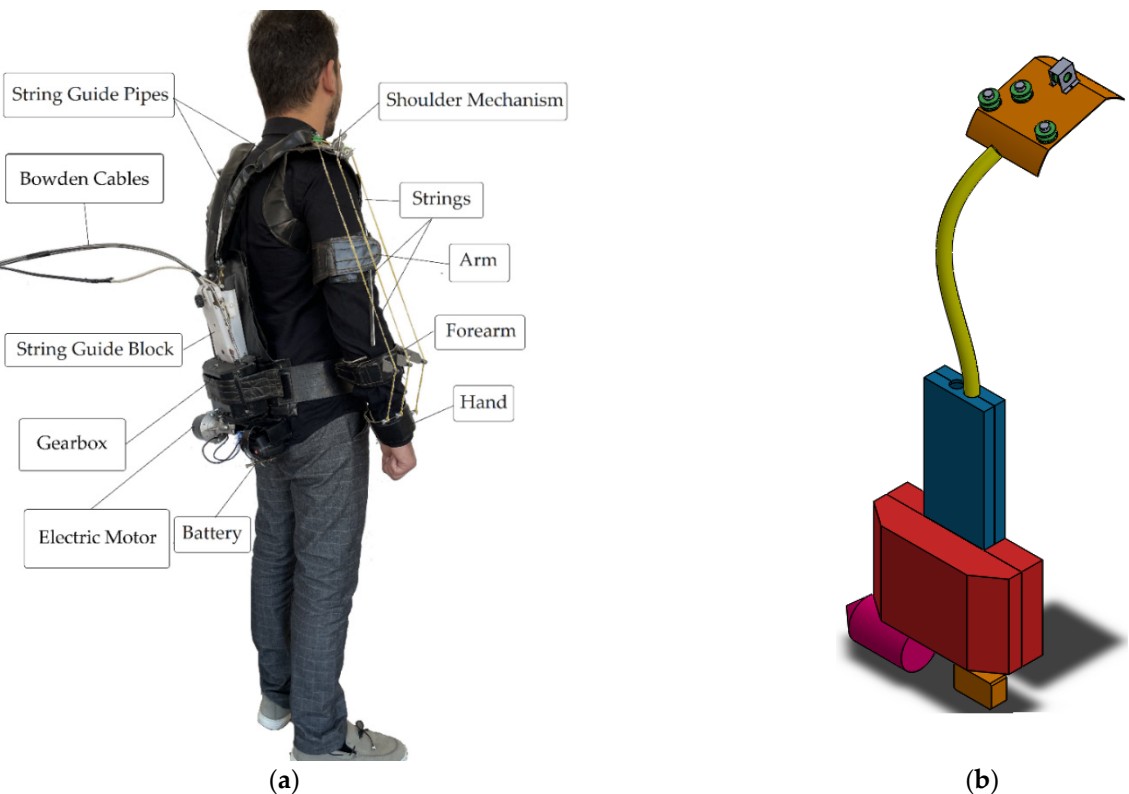

(**a**)            (**b**)

**Figure 9.** (**a**) The structure of the cable-driven exoskeleton of the upper limb, (**b**) CAD model.

For technical simplicity and testing convenience, instead of small electric motors actuating each string, one 30 W electric motor is used, which transmits the motion to each string via a gearbox (Figure 10a). The device is mechanically controlled (i.e., pulling the corresponding strings) by means of Bowden cables. The separation of strings and their tension process is regulated by means of pulleys placed in the string guide box. The strings are guided from the human back to the shoulder mechanism (Figure 10b) by the flexible string guide pipes. Then, they are attached to the arm, forearm and hand segments. The device works with a 12 V, 1.5 A small battery. The total weight of the device is 3.5 kg. Experiments show the effectiveness of the device and its compliance with the requirements of rehabilitation robotics.

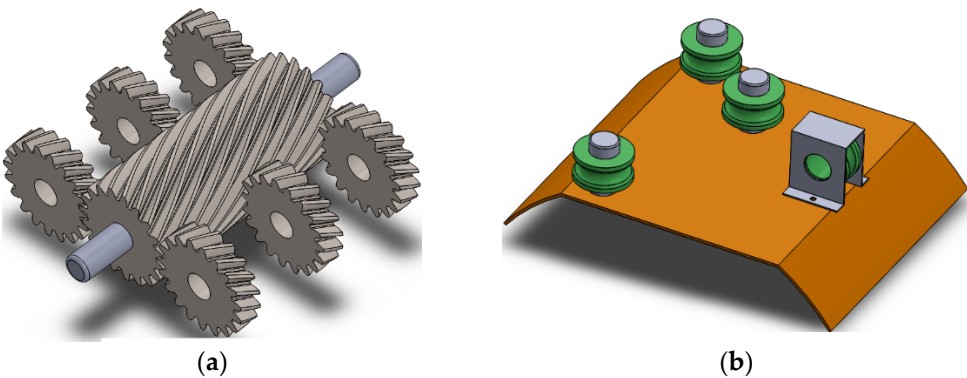

**Figure 10.** (**a**) Gearbox structure of the exoskeleton, (**b**) shoulder mechanism.

A more detailed description of the developed exoskeleton including the wearing and installation methods, as well as a long-term evaluation of the results during the rehabilitation treatment, are subjects of another large-scale study; we plan to present this in future studies.

## 7. Results

As a result of the optimization, the optimal weights of the exoskeleton segments, pulleys and electric motors, radii of pulleys, cable tensions, as well as cable connection angles or, in other words, the angles between tension lines and upper limb segments are determined (Table 3).

**Table 3.** The optimal weights of the exoskeleton segments, pulleys and electric motors, radii of pulleys, cable tensions, and cable connection angles.

| $m_{up}$ | $m_f$ | $m_h$ | $m_p$ | $m_e$ | $r_1$ | $r_2$ | $r_3$ | $r_4$ | $r_5$ |
|---|---|---|---|---|---|---|---|---|---|
| 2 | 1.1 | 0.2 | 0.1 | 0.4 | 0.07 | 0.05 | 0.02 | 0.02 | 0.03 |
| $F_1$ | $F'_1$ | $F_2$ | $F'_2$ | $F_3$ | $F'_3$ | $F_4$ | $F'_4$ | $F_5$ | $F'_5$ |
| 44.2 | 33.8 | 27.9 | 24.5 | 27.8 | 25.1 | 25.5 | 24.9 | 12.5 | 13.1 |
| $\sin \varphi_1$ | $\sin \varphi_2$ | $\sin \varphi_3$ | $\sin \varphi_4$ | $\sin \varphi_5$ | $\sin \varphi_6$ | $\sin \varphi_7$ | $\sin \varphi_8$ | $\sin \varphi_9$ | $\sin \varphi_{10}$ |
| 0.08 | 0.6 | 0.18 | 0.8 | 0.3 | 0.5 | 0.2 | 0.5 | 0.6 | 0.5 |

Now, let's discuss the effects of variable lever arms. For the simulation of human daily activities, we add a load of 5 kg to a user's hand. The following cable tension average values for the same lever arms were obtained (Table 4).

**Table 4.** Average values of cable tensions (additional loading).

| $F_1$ | $F'_1$ | $F_2$ | $F'_2$ | $F_3$ | $F'_3$ | $F_4$ | $F'_4$ | $F_5$ | $F'_5$ |
|---|---|---|---|---|---|---|---|---|---|
| 353 | 353 | 138 | 138 | 251 | 251 | 60 | 60 | 52 | 52 |

If we increase the lever arms by 30 mm, we get the following values of cable tensions (Table 5).

**Table 5.** Average values of cable tensions (increasing lever arms).

| $F_1$ | $F'_1$ | $F_2$ | $F'_2$ | $F_3$ | $F'_3$ | $F_4$ | $F'_4$ | $F_5$ | $F'_5$ |
|---|---|---|---|---|---|---|---|---|---|
| 150 | 150 | 67 | 67 | 75 | 75 | 41 | 41 | 32 | 32 |

Finally, the necessary forces developed by the Hill model artificial muscles and general deformations in the whole range of the simulation were obtained, which can be a guideline for other similar studies (Figure 11).

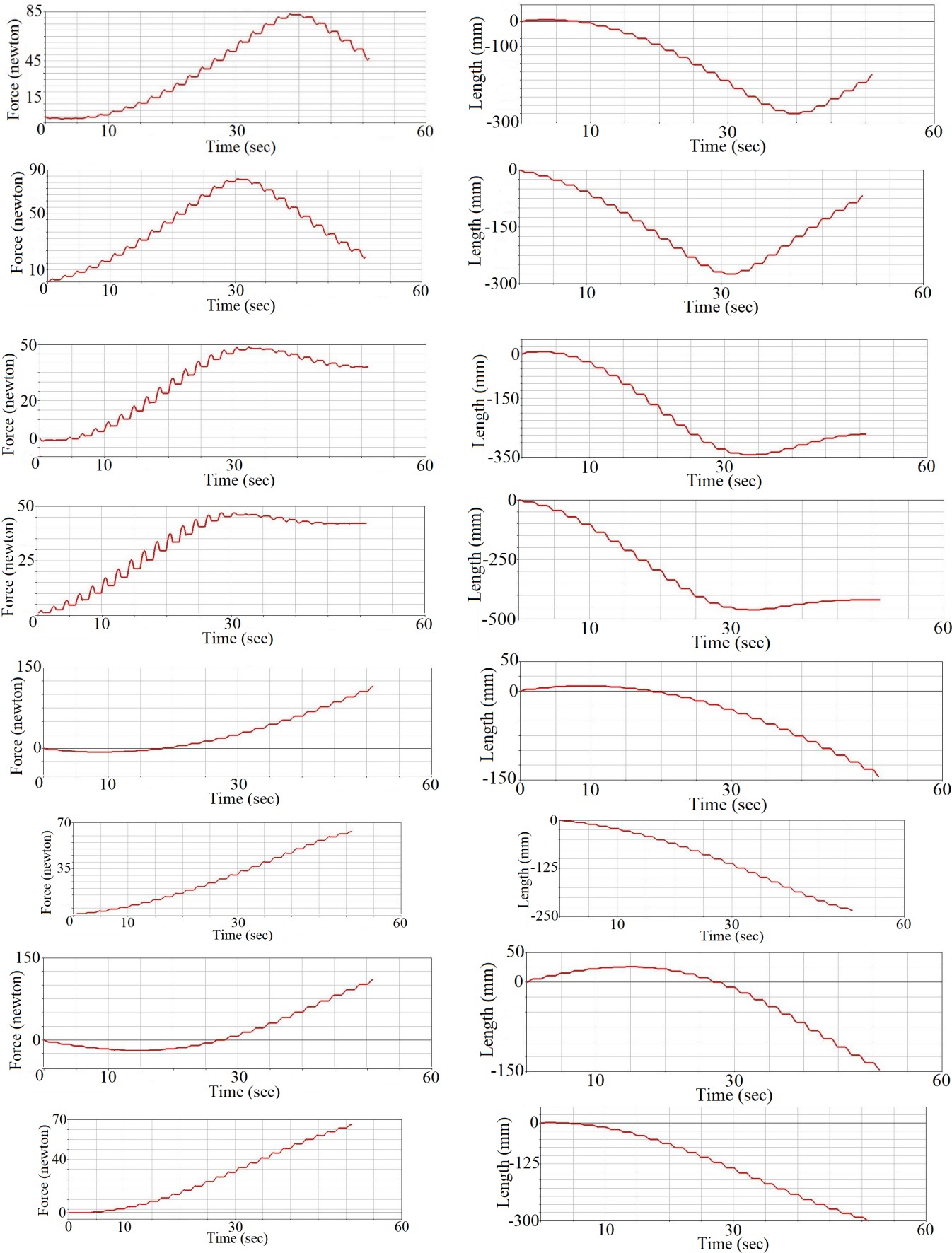

**Figure 11.** *Cont.*

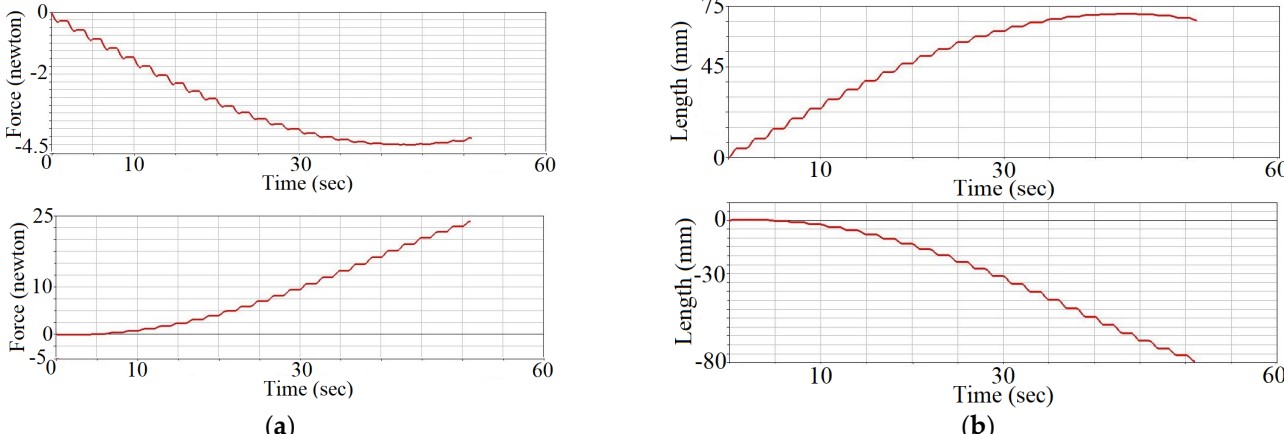

**Figure 11.** (**a**) The developed forces ($F_1, F_1', F_2, F_2', F_3, F_3', F_4, F_4', F_5, F_5'$), (**b**) general deformations of the Hill's model artificial muscles.

## 8. Discussion

In this paper, we have presented a bio-inspired mechanical design and control method for a new bio-inspired upper limb exoskeleton. Some of the main requirements to modern wearable exoskeletons: safety, stability, energy efficiency and versatility were achieved due to cable-driven over-actuation, EP control and variable torque generating mechanisms installed at the joints. Shift of the hand position during the whole motion was reached, and the relationship between the values of variable lever arms and cable tensions was revealed. The generalized exoskeleton model with artificial muscles has been proposed, taking into account Hill's model of natural muscles, which, as an "ideal" theoretical model, allows us to compare the results of the design of these types of exoskeletons and serves as a guide. The mathematical modeling and control of the proposed device were carried out by means of the MSC ADAMS system. As extensions of the present work, the following directions of further studies aimed to improve the main functional characteristics of the proposed bio-inspired exoskeleton can be outlined:

- A spatial model design, which will allow us to activate all degrees of freedom of upper limb, and consequently restore muscles functions.
- Nowadays, requirements of exoskeletons also include the ability to learn new skills, i.e., the creation of a so-called "smart" device is needed, which will greatly increase the efficiency of the device. This is again a good target for further follow-up studies.

**Supplementary Materials:** The following are available online at https://www.mdpi.com/article/10.3390/robotics10040123/s1, Video S1: manuscript-supplementary.MOV.

**Author Contributions:** N.Z. and M.H. established the methods of conceptual mechanical design of the exoskeleton, created Matlab algorithm for the design optimization and obtained numerical results, N.Z. established the methods of EP control, Y.S. established the methods of using artificial muscle model in the exoskeleton design. All authors contributed equally to the evaluation of the results and writing of this paper. All authors have read and agreed to the published version of the manuscript.

**Funding:** This research is funded by the Science Committee of the Ministry of Education and Science of the Republic of Armenia (Grant No. 21DP-2D006).

**Institutional Review Board Statement:** The study was conducted in accordance with the Declaration of Helsinki, and the protocol was approved by the Ethics Committee of robotics-1440822.

**Informed Consent Statement:** Informed consent was obtained from all subjects involved in the study.

**Data Availability Statement:** The data presented in this study are available on request from the corresponding author.

**Conflicts of Interest:** The Authors declare that there is no conflict of interest.

**Appendix A**

```
clear all; close all; clc
D = 30;
objf=inline('(−0.5*(x1 + 4 × x4 + 4 × x5) − x2 + 2 × x4 + 2 × x5 + x3)) × 9.8 × 0.3 × 0.17
− (x6 × x16/(0.1 × x21 + x16)) + (x7 × x16/(0.1 × x26 + x16)) − (x10 × x17/(0.15 × x23 + x17))
+ (x11 × x17/(0.15 × x23 + x17))','x1','x2','x3','x4','x5','x6','x7','x8','x9','x10','x11','x12','x13',
'x14','x15','x16','x17','x18','x19','x20','x21','x22','x23','x24','x25','x26','x27','x28','x29','x30');
objf = inline('(−0.5 × (x2 + 2 × x4 + 2 × x5 − x3) × 9.8 × 0.3 × 0.34 − (x8 × x18/(0.1 ×
x23 + x18)) + (x9 × x18/(0.1 × x28 + x18)) − (x10 × x17/(0.1 × x22 + x17)) + (x11 × x17/(0.1
× x27 + x17)) − (x14 × x19/(0.15 + 0.3) × x24 + x19)) + (x15 × x19/(0.15 + 0.3) × x29 +
x19))','x1','x2','x3','x4','x5','x6','x7','x8','x9','x10','x11','x12','x13','x14','x15','x16','x17','x18',
'x19','x20','x21','x22','x23','x24','x25','x26','x27','x28','x29','x30');
objf = inline('−0.5 × x3 × 9.8 × 0.1 × 0.5 − (x12 × x20/(0.1 × x25 + x20)) + (x13 × x20/(0.1
× x30 + x20)) − (x14 × x19/(0.4 × x24 + x19)) + (x15 × x19/(0.4 × x28 + x19))','x1','x2','x3','x4',
'x5','x6','x7','x8','x9','x10','x11','x12','x13','x14','x15','x16','x17','x18','x19','x20','x21','x22',
'x23','x24','x25','x26','x27','x28','x29','x30');
objf = inline('abs(x6 − x7),abs(x8 − x9),abs(x10 − x11),abs(x12 − x13),abs(x14 −
x15)','x1','x2','x3','x4','x5','x6','x7','x8','x9','x10','x11','x12','x13','x14','x15','x16','x17','x18',
'x19','x20','x21','x22','x23','x24','x25','x26','x27','x28','x29','x30');
objf = iline('x1 + x2 + x3 + 10 × x4 + 10 × x5','x1','x2','x3','x4','x5','x6','x7','x8','x9','x10',
'x11','x12','x13','x14','x15','x16','x17','x18','x19','x20','x21','x22','x23','x24','x25','x26','x27',
'x28','x29','x30');
objf = vectorize(objf);
N = 500;
Itmax = 100;
F = 1/30; CR = 0.8;
a(1:N,1) = 2; b(1:N,1) = 5; a(1:N,2) = 1; b(1:N,2) = 4; a(1:N,3) = 0.2; b(1:N,3) = 0.6;
a(1:N,4) = 0.1; b(1:N,4) = 0.5; a(1:N,5) = 0.5; b(1:N,5) = 1; a(1:N,6) = 1; b(1:N,6) = 50; a(1:N,7)
= 1; b(1:N,7) = 50; a(1:N,8) = 1; b(1:N,8) = 50; a(1:N,9) = 1; b(1:N,9) = 50; a(1:N,10) = 1;
b(1:N,10) = 50; a(1:N,11) = 1; b(1:N,11) = 50; a(1:N,12) = 1; b(1:N,12) = 50; a(1:N,13) = 1;
b(1:N,13) = 50; a(1:N,14) = 1; b(1:N,14) = 50; a(1:N,15) = 1; b(1:N,15) = 50; a(1:N,16) = 0.01;
b(1:N,16) = 0.1; a(1:N,17) = 0.01; b(1:N,17) = 0.1; a(1:N,18) = 0.01; b(1:N,18) = 0.1; a(1:N,19)
= 0.01; b(1:N,19) = 0.1; a(1:N,20) = 0.01; b(1:N,20) = 0.1; a(1:N,21) = 0.08; b(1:N,21) = 1;
a(1:N,22) = 0.08; b(1:N,22) = 1; a(1:N,23) = 0.08; b(1:N,23) = 1; a(1:N,24) = 0.08; b(1:N,24) = 1;
a(1:N,25) = 0.08; b(1:N,25) = 1; a(1:N,26) = 0.08; b(1:N,26) = 1; a(1:N,27) = 0.08; b(1:N,27) = 1;
a(1:N,28) = 0.08; b(1:N,28) = 1; a(1:N,29) = 0.08; b(1:N,29) = 1; a(1:N,30) = 0.08; b(1:N,30) = 1;
d = (b − a);
basemat = repmat(int16(linspace(1,N,N)),N,1);
basej = repmat(int16(linspace(1,D,D)),N,1);
x = a + d. × rand(N,D);
fx = objf(x(:,1),x(:,2),x(:,3),x(:,4),x(:,5),x(:,6),x(:,7),x(:,8),x(:,9),x(:,10),x(:,11),x(:,12),x(:,13),
x(:,14),x(:,15),x(:,16),x(:,17),x(:,18),x(:,19),x(:,20),x(:,21),x(:,22),x(:,23),x(:,24),x(:,25),x(:,26),x(:,27),
x(:,28),x(:,29),x(:,30));
[fxbest,ixbest] = min(fx);
xbest = x(ixbest,1:D);
for it = 1:itmax;
permat = bsxfun(@(x,y) x(randperm(y(1))),basemat',N(ones(N,1)))';
v(1:N,1:D) = repmat(xbest,N,1) + F × (x(permat(1:N,1),1:D) − x(permat(1:N,3),1:D));
r = repmat(randi([1 D],N,1),1,D);
muv = ((rand(N,D)<CR) + (basej = = r)) ~ = 0;
mux = 1 − muv;
u(1:N,1:D) = x(1:N,1:D). × mux(1:N,1:D) + v(1:N,1:D). × muv(1:N,1:D);
```

```
fu = objf(u(:,1),u(:,2),u(:,3),u(:,4),u(:,5),u(:,6),u(:,7),u(:,8),u(:,9),u(:,10),u(:,11),u(:,12),u(:,13),
u(:,14),u(:,15),u(:,16),u(:,17),u(:,18),u(:,19),u(:,20),u(:,21),u(:,22),u(:,23),u(:,24),u(:,25),u(:,26),
u(:,27),u(:,28),u(:,29),u(:,30)));
    idx = fu < fx;
    fx(idx) = fu(idx);
    x(idx,1:D) = u(idx,1:D);
    [fxbest,ixbest] = min(fx);
    xbest = x(ixbest,1:D);
    end
    [xbest,fxbest]
```

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
