# Peer review of "Bio-Inspired Conceptual Mechanical Design and Control of a New Human Upper Limb Exoskeleton"

_robotics, doi:10.3390/robotics10040123_

Round 1

Reviewer 1 Report

This paper completes a conceptual mechanical design of an upper limb exoskeleton, then uses differential evolution method to do design optimization, and analysises an equilibrium point control (EP) method for the exoskeleton. A theoretical model of the “ideal” artificial muscle exoskeleton is also developed using Hill's natural muscle model. The structure and content of this article are easy to follow, and this project is well-prepared; however, there are several comments and questionable points on the work:

  1. The conceptual mechanical design of the exoskeleton is not clear. The wearing and installation methods of the exoskeleton are not mentioned, only the mathematical modeling of the exoskeleton is completed.
  2. It’s incomplete for the description of the exoskeleton design optimization. It only introduces the optimization algorithm used but does not show the actual optimization effect and the guiding role for the mechanical structure design.
  3. For the experimental validation, from the Figure 9. a), the experimental test platform used only includes the drive module part, but does not include the exoskeleton structure designed in the previous article. And there is no description of how the test experiment was implemented.
  4. The description of the experiment preparation and final results in Sections 6 and 7 of the article is relatively brief, so there are doubts about the rationality of the experimental results.

Author Response

Dear reviewer, 
thank you for your revision, it was very helpful!

Reviewer 2 Report

This paper presents the mechanical design and control of a bio-inspired upper limb exoskeleton, both modeling, simulation and experiment are provided. The questions are given as follows:

  1. The quality of all figures are very poor, it is difficult for me to evaluate these figures. The author must provide new figures with high quality.
  2. What is the main design and control purpose of the mechanism?
  3. How to evaluate the experimental results? The author should tell us the main focused index, and more analyses are needed.
  4. The conclusion part is needed.
  5. 5. What is the main advantages of the mechanism compared to other existing upper limb exoskeletons?

Author Response

(The authors gave the same response as above.)

Reviewer 3 Report

The paper proves to be an interesting read and the theme covers a relevant field that is of great value to the advancement of medical robotics.

However, there are some critical aspects that need to be resolved in order for the paper to be publish.

Minor observations:

- Please reformulate the sentence that include line 78 “ensured which ensured..”;

- Typo correction on line 87 “redunduncy”;

- Sentence from line 94 needs to be reformulate “...angle between the upper and lower limb decreases”; (arm and forearm)

- Please clarify the meaning of the line 101 “Bi-articular cables”.

- What is the difference between element 5 and 6 in figure 3? A higher resolution image might be useful, since the lines that indicate the element number for 5 and 6 are hard to be distinguish.

- The authors should take into consideration the following papers as inspiration for improvements:

10.1109/ICAR.1991.240572

10.1109/ICRA.2014.6907024

10.3390/sym12091470

10.3390/s21165411

10.3390/app11135865

10.1109/IROS.2006.281712

10.1109/ICMA.2005.1626539

10.1109/IROS.2015.7354223

10.1109/MRA.2014.2360283

Critical observations:

  1. I suggest to explain in more detail the operation of the drive system shown in Figure 3, i.e., to be clear who is operating / wherefrom the elements 2 are actuated (if I understood well, they are not actuators, but actuated elements - cables).
  2. If the cable 2 from the upper part of the actuating mechanism is tensioned and exceeds a certain threshold value, the upper lever 3 will not rotate counter-clockwise as specified in the paper but clockwise. In this case it must be specified what kind of screw-nut mechanism is the one in question (on the right or on the left-hand side) to increase the lever arm (as shown in the drawing must be one on the left-hand side). It could be on the right-hand side, if the cable were wound on lever 3 in the opposite direction.

The statement made by the authors in the text is valid for the lower part of the actuating mechanism (the levers start to rotate in the anti-clockwise direction, so, the lever arms of the cable tensile forces are increasing, if the screw-nut mechanism is a right-hand side one).

  1. In chapter 3 of the paper optimizations are discussed. Before talking about optimizations first you need to have an estimation of the parameters that need to be optimized. Here there are two options:
  2. a) Taking this into consideration, it is suggested the authors start by mentioning the methods used for determining the mass before optimization is applied (without detailing how this was done).
  3. b) Normally there needs to be an initial design that can be used in a CAD environment to determine the mass of the assembly. The other way of determining the mass is by building the device and measuring the mass. None of the two methods are covered before applying the optimization.
  4. An engineering workflow for developing such a device is not present. Optimizations on the mass of the device can’t be discussed without having performed a rigorous finite element analysis. This finite element analysis takes into consideration the stress of critical components under static and dynamic load. From this analysis you can determine if the parts in the design will or will not break during operation. The geometry (design) of the parts is also critical for optimizations. But this idea could be introduced in some future researches.
  5. Reducing the cable tension is a good idea to optimize the device, but there are other parameters that can be very detrimental to cable actuated devices. One parameter that is not considered properly is friction. For example, the friction between parts 3 and 4 could cause the device to block during operation. Another important aspect that is not taken into consideration is the friction of the Bowden cable transmission. I suggest that aspects to be mentioned only, because the article is consistent enough in terms of scientific achievements.
  6. Higher torque from higher cable tension does not necessarily indicate the need of a bigger motor. Gear boxes are usually used and calculated accordingly. This aspect should be taken into account in the presented paper.
  7. In figure 10 the gearbox and the shoulder mechanism are not explained sufficiently. It is not clear how it works.

Additionally, I suggest that it would be of great value if the authors would provide videos or more pictures of the testing via the MDPI platform.

Author Response

(The authors gave the same response as above.)

Round 2

Reviewer 2 Report

The paper can be accepted.